# The Effect of Exogenous Nitrate on LCO Signalling, Cytokinin Accumulation, and Nodule Initiation in *Medicago truncatula*

**DOI:** 10.3390/genes12070988

**Published:** 2021-06-28

**Authors:** Kerstin Gühl, Rens Holmer, Ting Ting Xiao, Defeng Shen, Titis A. K. Wardhani, René Geurts, Arjan van Zeijl, Wouter Kohlen

**Affiliations:** 1Laboratory of Molecular Biology, Department of Plant Sciences, Wageningen University & Research, 6708 PB Wageningen, The Netherlands; kerstin.guhl@gmail.com (K.G.); Rens.holmer@wur.nl (R.H.); tintin.xiao@gmail.com (T.T.X.); dshen@mpipz.mpg.de (D.S.); titis.wardhani@wur.nl (T.A.K.W.); rene.geurts@wur.nl (R.G.); arjan.vanzeijl.molbi@gmail.com (A.v.Z.); 2Bioinformatics Group, Department of Plant Sciences, Wageningen University, 6708 PB Wageningen, The Netherlands

**Keywords:** *Medicago truncatula*, nitrate, nodulation, ethylene, cytokinin

## Abstract

Nitrogen fixation by rhizobia is a highly energy-demanding process. Therefore, nodule initiation in legumes is tightly regulated. Environmental nitrate is a potent inhibitor of nodulation. However, the precise mechanism by which this agent (co)regulates the inhibition of nodulation is not fully understood. Here, we demonstrate that in *Medicago truncatula* the lipo-chitooligosaccharide-induced accumulation of cytokinins is reduced in response to the application of exogenous nitrate. Under permissive nitrate conditions, perception of rhizobia-secreted signalling molecules leads to an increase in the level of four cytokinins (i.e., iP, iPR, tZ, and tZR). However, under high-nitrate conditions, this increase in cytokinins is reduced. The ethylene-insensitive mutant *Mtein2*/*sickle*, as well as wild-type plants grown in the presence of the ethylene biosynthesis inhibitor 2-aminoethoxyvinyl glycine (AVG), is resistant to the inhibition of nodulation by nitrate. This demonstrates that ethylene biosynthesis and perception are required to inhibit nodule organogenesis under high-nitrate conditions.

## 1. Introduction

Most legumes can engage in a nitrogen-fixing endosymbiosis with diverse bacterial species collectively known as rhizobia [1]. In this interaction, rhizobia are accommodated intracellularly in specialized root organs called nodules. These nodules create optimal physiological conditions for rhizobia to convert atmospheric nitrogen into ammonium using the nitrogenase enzyme complex. Fixation of atmospheric nitrogen by nitrogenase is energetically costly and, therefore, legume nodule formation is strictly regulated based on both internal and external cues [1]. Understanding this regulation will be instrumental to successfully expanding the agricultural application of rhizobium nitrogen fixation beyond legumes [2]. 

Although there are exceptions [3,4], legume–rhizobium symbiosis is initiated upon perception of rhizobium-secreted lipo-chitooligosaccharide (LCO) signalling molecules [5]. Perception of these external signals results in the activation of a suite of transcriptional regulators, including the master regulator of symbiotic signalling NODULE INCEPTION (NIN) [6,7,8,9], and accumulation of the bioactive cytokinins isopentenyl adenine (iP) and *trans*-zeatin (tZ) [6]. Both induction of *NIN* and activation of cytokinin signalling are required in order to mitotically reactivate cells in the pericycle and root cortex [9,10,11,12,13,14], resulting in the formation of a nodule primordium [15,16]. Simultaneously, LCO perception initiates a bacterial infection process that starts in the root hair cell that perceives the LCO signal. The bacterial infection process involves a redirection of root hair growth, resulting in the formation of a tight curl that entraps the bacteria in an infection pocket [17]. From there, cell-wall-bound infection threads are formed that guide the bacteria to the developing nodule primordium. Once inside the nodule, rhizobia are released into the host cell cytoplasm as transient, nitrogen-fixing, organelle-like structures. These so-called symbiosomes remain surrounded by a host-derived membrane, which facilitates the exchange of ammonium and photoassimilates between the host plant and the microbes [18].

In most legumes, nodule initiation is negatively regulated by exogenous sources of fixed nitrogen [19,20,21]. This inhibitory effect appears to be gradual, and depends on both the concentration [19] and the source of fixed nitrogen applied to the plant [22,23]. Partial resistance to nitrogen inhibition of nodulation is observed in alfalfa (*Medicago sativa*) plants when treated with the ethylene biosynthesis inhibitor 2-aminoethoxyvinyl glycine (AVG) [24]. Furthermore, it was shown that the release of ethylene from inoculated alfalfa roots is increased after the addition of nitrates [25]. Taken together, this suggests that ethylene could function as a signalling intermediate during inhibition of nodulation by fixed forms of nitrogen. 

In contrast to the positive role for cytokinin during nodule initiation, ethylene has been identified as a negative regulator of nodulation [26,27]. Ethylene is produced through consecutive activity of 1-aminocyclopropane-1-carboxylic acid (ACC) synthases (ACS) and ACC oxidases (ACO) [28]. The conversion of *S*-adenosylmethionine into ACC through ACS activity is generally considered to be rate limiting [28]. Several ACS-encoding genes in *M. truncatula* are transcriptionally induced in response to LCO signalling [6,7,29], and an increase in ethylene release has been reported for *Lotus japonicus* at 6 h after inoculation [30]. The *sickle* mutant in *M.*
*truncatula* is ethylene insensitive due to a mutation in *EIN2*, which encodes a central component in the ethylene signalling cascade [26,27,31]. Additionally, the *sickle* mutant is hyperinfected by rhizobium, and forms a multitude of nodules in distinct clusters along the root [26]. A similar phenotype is observed in *L. japonicus* when both EIN2-encoding genes are mutated [30,32]. Contradicting reports describe the effect of the *sickle* mutation on the inhibition of nodulation by nitrate. Prayitno [33] demonstrated that nodulation on *sickle* is less affected by elevated levels of nitrate, whereas others reported nitrate inhibition of nodulation in *sickle* to be identical to that in wild-type plants [34,35]. Therefore, the contribution of ethylene to the inhibition of nodulation by nitrate remains unclear. 

Here, we set out to investigate the role of cytokinin and ethylene in the inhibition of nodulation by nitrate in *M.*
*truncatula*. We demonstrate that the application of 16 mM NO_3_^−^ completely blocks the initiation of pericycle and cortical cell divisions associated with the formation of nodule primordia. We show that, at this concentration of nitrate, LCO signalling is not affected, whereas the accumulation of cytokinin in response to LCOs is reduced, though not entirely inhibited. In addition, our data indicate that ethylene biosynthesis and signalling are required for the inhibition of pericycle and cortical cell divisions observed at high nitrate concentrations. 

## 2. Materials and Methods

### 2.1. Plant Material and Growth Conditions

*M.**truncatula* seeds (Jemalong A17 wild type, *Mtein2*/*sickle* [26]) were treated with H_2_SO_4_ for 7 min, rinsed 5 times with Milli-Q water, and sterilized for 10 min using normal household bleach. Seeds were washed with sterile Milli-Q water 5 times and placed on round petri dishes containing Fåhraeus medium [36] (0.25 mM NO_3_^−^, 1% Daishin agar) at 4 °C in darkness for stratification. After 48 h, seeds were transferred to room temperature to germinate for an additional 24 h. Germinated seedlings were transferred to the appropriate in vitro growth system. For all experiments, plants were grown in an environmentally controlled growth chamber at 20 °C/18 °C with a 16 h-light/8 h-dark cycle and 70% relative humidity.

### 2.2. Testing the Effects of Varying Nitrate Concentration on Nodulation

A modified Fåhraeus medium (0.12 g/L MgSO_4_·7H_2_O, 0.10 g/L KH_2_PO_4_, 0.15 g/L Na_2_HPO_4_·2H_2_O, 1 mL/L 15 mM Fe-Citrate, 2.50 mL/L Spore elements β-(CuSO_4_·5H_2_O 0.0354 g/L, MnSO_4_·H_2_O 0.462 g/L, ZnSO_4_·7H_2_O 0.974 g/L, H_3_BO_3_ 1.269 g/L, Na_2_MoO_4_·2H_2_O 0.398 g/L), pH 6.7) [36] with varied levels of nitrate (0, 0.125, 0.25, 0.5, 1, 2, 4, and 8 mL 1 M Ca(NO_3_)_2_) was used. Increasing Ca^2+^ levels were compensated by adding decreasing amounts (8.7, 8.575, 8.45, 8.2, 7.7, 6.7, 4.7, and 0.7 mL) of 1 M CaCl_2_. For root hair deformation, individual seedlings were transferred to Fåhraeus slides [36]. For the plate growth assay, medium containing 1% Daishin agar was used. Nodulation was scored on square (12 cm × 12 cm) plates with different nitrate concentrations. Eight seedlings were grown per plate, and a minimum of four plates per replicate were grown. Plates were partially covered with tin foil to shade roots. All plants were grown under long-day conditions (16 h of light 21 °C and 8 h of darkness 18 °C). From each plate in the assay, susceptible zones (± 1 cm) of ca. 7–10 roots were treated with Nod factors, collected and snap-frozen in liquid nitrogen. Samples were stored at −80 °C before further processing (i.e., to determine gene expression or cytokinin levels).

### 2.3. Application of Nod Factors and Rhizobia

Nod factors (*Sm2011NF*) from *Sinorhizobium*
*meliloti* strain 2011 were purified and applied as previously described [6,37]. *Sn2011NF* stocks were stored in 100% DMSO and diluted 100-fold (~10^−9^ M) in Fåhraeus medium [36]. As a mock treatment, Fåhraeus medium (1% DMSO) was used. Nitrate levels used during seedling growth before inoculation were maintained during inoculation. *Sn2011NF* was applied to the root susceptible zone. Roots were exposed for 3 h and, subsequently, 1-cm root pieces were cut just above the root tip and snap-frozen in liquid nitrogen (*n* = 6 for cytokinin measurements, *n* = 3 for gene expression analysis). For rhizobia application, *S. meliloti* strain 2011 was grown and diluted to an OD_600_ of 0.01 for direct application on plates or inoculation in our semisterile growth system. 

### 2.4. RNA Isolation, cDNA Synthesis, and Quantitative RT-PCR

RNA was isolated from snap-frozen root samples using the plant RNA kit (E.Z.N.A, Omega Biotek, Norcross, GA, USA) according to the manufacturer’s protocol. Then, 1 μg of total RNA was used to synthesize cDNA, using the iScript cDNA synthesis kit (Bio-Rad, Hercules, CA, USA) according to the manufacturer’s protocol. For real-time qRT-PCR, 10-µL reactions with 2× iQ SYBR Green Super-Mix (Bio-Rad, Hercules, CA, USA) were run on a CFX Connect optical cycler (Bio-Rad, Hercules, CA, USA), according to the manufacturer’s protocol. All primers, including the genes used for normalization (*MtUBQ10* and *MtPTB*), are detailed in Appendix A. Data analysis was performed using CFX Manager 3.0 software (Bio-Rad, Hercules, CA, USA). Cq values of 32 and higher were excluded from the analysis, though still checked for transcriptional induction. Statistical significance was determined based on Student’s *t*-test (*p* < 0.01).

### 2.5. Cytokinin Extraction

For cytokinin extraction from *M. truncatula* material, ~20 mg of snap-frozen root and nodule material were used per sample. The tissue was ground to a fine powder at −80 °C using 3-mm stainless steel beads at 50 Hz for 1 min in a TissueLyser LT (Qiagen, Germantown, USA). Ground samples were extracted with 1 mL of 100% methanol (MeOH) containing stable isotope-labelled internal standards (IS, Appendix A). Internal standards were used at an end concentration of 100 nM per compound per sample. Samples were vortexed, ultrasonicated for 30 s, and extracted at 4 °C overnight on an orbital shaker. Subsequently, samples were centrifuged at 12,000 rpm for 10 min in a tabletop centrifuge set at 4 °C. Supernatants were transferred to amber 4-mL glass vials. Pellets were re-extracted with 1 mL of 100% MeOH for 1 h at 4 °C. After centrifugation, as described above, both supernatants were pooled before being evaporated to dryness in a speed vacuum system (SPD121P, ThermoSavant, Hastings, UK). Pellets were eluted in 1 mL of 1 M formic acid (in water) and loaded onto a 30 mg Oasis MCX Cartridge (Waters, Milford, OH, USA). Prior to sample loading, each cartridge was washed with 1 mL of MeOH and equilibrated with 1 mL of 1 M formic acid (water). After loading, the cartridge was washed with 1 mL of 0.35 M NH_4_OH (in water), and then eluted with 1 mL of 0.35 M NH_4_OH (in 60% MeOH). The 0.35 M NH_4_OH (in 60% MeOH) was evaporated in a speed vacuum system (SPD121P, ThermoSavant, Hastings, UK) at RT, and the residue stored at −20 °C until further analysis.

### 2.6. Detection and Quantification of Cytokinins via Liquid Chromatography–Tandem Mass Spectrometry

Samples were resuspended in 100 µL of methanol/water (0.1% formic acid) (10:90, *v*/*v*) and filtered through a 0.45 mm Minisart SRP4 filter (Sartorius, Goettingen, Germany). Analysis of cytokinins from *M. truncatula* was performed by comparing retention times and mass transitions with those of unlabelled standards (Appendix A), using a Waters XevoTQs mass spectrometer equipped with an electrospray ionization source coupled with an Acquity UPLC system (Waters, Milford, OH, USA). Chromatographic separations were conducted using an Acquity UPLC BEH C18 column (100 mm, 2.1 mm, 1.7 mm; Waters, Milford, OH, USA) by applying a methanol/water (0.1% formic acid) gradient. The column was operated at 40 °C with a flow rate of 0.25 mL·min^−1^, and was equilibrated for 30 min using a methanol/water (0.1% formic acid) (5:95, *v*/*v*) composition at the start of the run. The methanol/water (0.1% formic acid) gradient started from 5% (*v*/*v*) methanol, increasing to 70% (*v*/*v*) methanol in 17 min. To wash the column, the water/methanol gradient was increased to 100% methanol in a 1.0 min gradient, then maintained for 1.0 min before returning to 5% methanol using a 1.0 min gradient, prior to the next run. The sample injection volume was set to 5 µL. The mass spectrometer was operated in positive electrospray ionization mode when required. Cone and desolvation gas flows were set to 150 and 800 L·h^−1^, respectively. The capillary voltage was set at 3.0 kV, the source temperature at 150 °C, and the desolvation temperature at 550 °C. The cone voltage was optimized for each standard compound using the IntelliStart MS Console (Waters, Milford, OH, USA). Argon was used for fragmentation by collision-induced dissociation. Multiple reaction monitoring (MRM) was used for quantification [6,38,39]. Parent–daughter transitions for the different (stable isotope labelled) compounds were set using the IntelliStart MS Console. MRM transitions selected for compound identification and quantification are shown in Appendix A. The cone voltage was set to 40 eV. To determine sample concentrations, a 10-point calibration curve was constructed for each compound ranging from 0.1 µM to 19 pM, in addition to a known amount of an appropriate deuterium-labelled internal standard.

### 2.7. ACC Extraction, Detection, and Quantification via Liquid Chromatography–Tandem Mass Spectrometry

ACC analysis from the *M.*
*truncatula* root was performed as previously described [40]. 

### 2.8. Statistical Analysis

*NIN* gene expression was modelled with a generalized additive model (GAM) with smoothing splines [41]. The Akaike information criterion was used to determine that both NO_3_^−^ and LCO are relevant for explaining *NIN* expression (Appendix A). Hormone concentrations were modelled using a linear model of the form $concentration\;\sim \;NO^{3}+LCO+NO^{3}*LCO$, of which the significance of individual terms and the interaction effect was evaluated with an *F*-test (Appendix A). All models were fitted in R [42]; code and data are available at http://github.com/holmrenser/nitrate_cytokinin_nodulation (accessed on 15 May 2021). All other data were subjected to either Student’s *t*-test (Microsoft Excel, asterisk) or a one-way ANOVA (letters). Individual differences were then identified using Tukey’s post-hoc test, using SAS_9.20 (*p* < 0.05).

## 3. Results

### 3.1. Elevated Nitrate Concentrations Inhibit Nodulation in a Synchronized In Vitro Assay

To study the inhibition of nodulation by nitrate, we first determined at what nitrate concentration nodulation is blocked in *M.*
*truncatula*, and at which point in the LCO signalling cascade nitrate interferes. Previous results on this were obtained using a variety of non-synchronized experimental systems that often include the ethylene biosynthesis inhibitor AVG [19,20,21]. Therefore, we first implemented a synchronized in vitro nodulation assay independent of AVG application (Appendix A). For this, seedlings of *M.*
*truncatula* were grown on Fåhraeus medium with increasing concentrations of Ca(NO_3_)_2_, resulting in an exogenous nitrate concentration ranging from 0 to 16 mM NO_3_^−^. The concurrent increase in Ca^2+^ concentration was compensated for by the addition of CaCl_2_ (Appendix A). As light exposure is known to trigger ethylene biosynthesis in roots, and ethylene is a potent suppressor of nodulation, we shaded the lower section of our plates with tin foil. This allowed efficient nodule formation without the addition of AVG (Appendix A).

The root length was predetermined prior to rhizobia application. No effect of nitrate was observed at this timepoint (five days after germination (DAG)) (Appendix A), indicating that the varying concentrations of nitrate did not affect primary root growth and, thus, plant fitness. Next, plants were inoculated with *S.*
*meliloti* strain 2011, and nodule numbers were scored at 7 and 14 days post-inoculation (DPI) (Figure 1A, Appendix A). At 7 DPI, nodules developed normally on plants grown at 0, 0.25, and 0.5 mM NO_3_^−^. On plants grown at 1 and 2 mM NO_3_^−^, nodule numbers were reduced, and on plants grown at 4, 8, and 16 mM NO_3_^−^, nodules were not observed (Figure 1A). A similar pattern emerged at 14 DPI, suggesting that the inhibitory effect of nitrate is stable over time, although some nodules were observed on plants grown at 4 mM NO_3_^−^ at 14 DPI (Appendix A). In addition to reduced nodule numbers, whereas nodules formed on plants grown at 0, 0.25 and 0.5 mM NO_3_^−^ were the usual pink, nodules formed on plants grown at 1 and 2 mM NO_3_^−^ remained white. Sectioning of plastic-embedded nodules revealed that nodules formed on plants grown at 0.25 mM NO_3_^−^ developed normally, and displayed a zonation characteristic of indeterminate nodules (meristem, infection zone, and fixation zone; Figure 1B). Nodules formed on plants grown at 2 mM NO_3_^−^ also developed a meristem and infection zone; however, a stable fixation zone was not observed. Instead, infected cells seemed to senesce almost instantly (Figure 1C).

### 3.2. Root Hair Deformation Is Not Affected by Exogenous Nitrate

To systematically determine at which stage nitrate affects nodulation, we first examined whether nitrate interferes with root hair deformation. To test this, we transferred 1-day old seedlings to Fåhraeus slides [36] and grew them for 2 days on different concentrations of nitrate. Treatment with ~10^−9^ M LCOs from *S. meliloti* strain 2011 for 3 h induced root hair curling on all tested nitrate concentrations, with no apparent differences between concentrations. This indicates that, in our setup, root hair curling is not susceptible to inhibition by nitrate (Appendix A). 

### 3.3. Pericycle and Cortical Cell Divisions Are Blocked by High Nitrate

Next, we tested whether early induction of LCO-signalling genes is affected by nitrate. For this, we focused on *NODULE INCEPTION* (*NIN*). Plants were grown in our in vitro assay without AVG. Under control conditions, *NIN* expression is upregulated within 3 h of LCO application (Figure 2A), which is consistent with previous reports [6]. Interestingly, *NIN* expression was also induced across the entire nitrate concentration range after 3 h of LCO application, although its induction was most prominent in plants grown on 0.5 and 1 mM of nitrate (Figure 2A, Appendix A).

Cytokinins are known regulators of cell division, and it was previously demonstrated that LCO signalling triggers rapid accumulation of cytokinins in the region of the *M.*
*truncatula* root susceptible to nodulation—called the root susceptible zone [6]. In line with these previous observations, under permissive nitrate conditions, iP and tZ accumulated within 3 h of LCO application (Figure 2B,C, Appendix A). In addition, their corresponding riboside forms (iPR and tZR, respectively) showed a similar accumulation under these permissive nitrate conditions (Appendix A, Appendix A). Although *cis*-zeatin (cZ) and *cis*-zeatin riboside (cZR) were present in all samples, their levels were not affected by LCO application (Appendix A, Appendix A). To determine whether this response is affected by nitrate availability, we measured cytokinin concentrations at 3 h after LCO exposure in plants grown on increasing levels of nitrate. In mock-treated samples, nitrate concentrations did not affect the level of any of the analysed cytokinins (Figure 2B,C, Appendix A, Appendix A). Furthermore, increasing levels of nitrate in combination with LCO application did not affect the levels of cZ or cZR (Appendix A, Appendix A). However, elevated nitrate reduced the accumulation of tZ and tZR upon LCO application (Figure 2B,C, Appendix A, Appendix A). This was not the case for iP and iPR, whose levels did increase upon LCO treatment, but this induction did not vary significantly across the nitrate concentrations (Figure 2B,C, Appendix A, Appendix A). Combined, this shows that nitrate interferes with the LCO-induced accumulation of cytokinin in the *M.*
*truncatula* root susceptible zone. 

As both (partial) induction by LCO signalling of *NIN* and cytokinin accumulation still occur at high nitrate concentrations, we questioned whether the initiation of pericycle and cortical cell divisions occurs under these conditions. To this end, the root susceptible zone of plate-grown *M.*
*truncatula* seedlings was spot-inoculated with 10 µL of *S. meliloti* strain 2011 (OD_600_ 0.01) and sectioned at 4 DPI. This showed that cell divisions were initiated at all tested concentrations (Figure 3A–G), except at 16 mM (Figure 3H). Although cell divisions were initiated at 4 and 8 mM NO_3_^−^, well-developed nodule primordia could not be observed (Figure 3F,G). Additionally, the percentage of roots initiating cell division at 4 and 8 mM NO_3_^−^ was reduced by ~50% (Figure 3I). Altogether, this suggests that increasing levels of nitrate gradually reduce the ability to induce and sustain the cortical cell division associated with nodule primordium formation. 

### 3.4. Nitrate Inhibition of Pericycle and Cortical Cell Divisions Is Mediated By Ethylene

Ethylene is a potent inhibitor of pericycle and cortical cell divisions [43,44]. Moreover, in peas (*Pisum sativum*), the ethylene biosynthesis gene *ACC OXIDASE* is expressed opposite phloem poles and, therefore, antagonistically correlates with the position where nodule primordia are initiated [45]. To determine whether ethylene might also be involved in the inhibition of cell divisions during the inhibition of nodulation by nitrate, we studied the nodulation behaviour of the ethylene-insensitive *sickle* mutant. Consistent with previous reports, the *sickle* mutant forms about eight times more nodules than wild type at low nitrate concentrations (Figure 4A; [26]). At 2 mM NO_3_^−^, the average nodule number is reduced by ~50% on *sickle* mutant roots compared to that at 0.25 mM NO_3_^−^, similar to the reduction observed on wild-type roots (Figure 4A–C). Interestingly, increasing concentrations of nitrate do not further reduce nodule numbers (Figure 4A). Instead, a comparable number of nodules forms on *sickle* plants grown at nitrate concentrations ranging from 2 to 16 mM NO_3_^−^ (Figure 4A), confirming observations by Prayitno [33]. Consistently, cytokinin measurements after LCO exposure indicate a strong accumulation of iP and tZ in the roots of the *sickle* mutant grown at both 0.25 and 16 mM NO_3_^−^ (Figure 5A–D). At 0.25 mM NO_3_^−^, iP and tZ concentrations increase roughly 12 and 100 times, respectively, after LCO treatment. At 16 mM NO_3_^−^, this response is reduced by 50% (Figure 5B,D), but still far exceeds the response in wild-type roots grown at 0.25 mM NO_3_^−^ (Figure 5A,C). 

To determine whether a reduction in ethylene biosynthesis similarly confers nitrate resistance in *M. truncatula*, we grew seedlings on Fåhraeus medium containing 2 µM AVG. This reduced ACC concentrations in the root susceptible zone by ~50% (Figure 6A). At 0.25 mM NO_3_^−^, AVG treatment increased nodule numbers by 30–50%. At 16 mM NO_3_^−^, nodules were formed on plants treated with AVG, but not on mock-treated plants (Figure 6B). This indicates that a reduction in ACC content creates resistance to nitrate. Combined, these results indicate that ethylene biosynthesis and perception are both required for the inhibition of nodulation by nitrate.

## 4. Discussion

Here, we investigated the effects of nitrate on LCO signalling and subsequent nodule formation in *M.*
*truncatula*. We found a concentration-dependent effect of nitrate on LCO-induced accumulation of tZ and tZR. The ethylene-insensitive mutant *Mtein2*/*sickle* and AVG-grown plants both display reduced sensitivity to nitrate during nodulation. This suggests that, in *M.*
*truncatula*, ethylene is involved in the inhibition of nodule initiation by nitrate.

### 4.1. Nitrate Interferes with Both Nodule Initiation and Nitrogen-Fixation Rates

We adopted a plate system that allows efficient nodulation without the addition of chemical inhibitors (e.g., AVG). In this system, the inhibitory effect of nitrate on nodulation seems to be a two-step process. Intermediate nitrate concentrations (1 and 2 mM NO_3_^−^) reduce nodule numbers by 50%. Nevertheless, the nodules that do form are white, and do not contain a well-defined fixation zone. This suggests that these nodules are most likely non-fixing [46,47]. At higher concentrations of nitrate (≥4 mM NO_3_^−^), nodules are only occasionally observed. 

Taken together, this suggests that nitrate interferes with nodulation on at least two different levels: high nitrate concentration (≥4 mM NO_3_^−^) leads to a strong reduction in nodule numbers by interfering with cytokinin-induced pericycle and cortical cell divisions, whereas medial nitrate levels (1–2 mM NO_3_^−^) reduce the symbiotic effectiveness of developing nodules. A recent study on the common bean (*Phaseolus vulgaris*) painted a similar picture [48]; with a nodulation experiment over a nitrate concentration range, the authors demonstrated that although medial levels of exogenous nitrate lead to increased numbers of nodules, the rate of N_2_ fixation already drops at 2.5 mM NO_3_^−^. This dual effect of nitrate is consistent with an earlier report that also described a higher sensitivity to nitrate with regard to nitrogen-fixation rates compared to nodule formation [19]. Additionally, high levels of exogenous nitrate lead to nodule senescence [49,50,51,52]. It is possible that the mechanisms controlling nitrate-induced nodule senescence are also at work here. Legumes might have evolved such a dual sensitivity to nitrate in order to cope with non-uniformly distributed soil nitrogen concentrations. Blocking nodulation completely when roots pass through a nitrogen-rich patch might be unfavourable, as it could limit the plants’ adaptability to changing soil nitrogen concentrations. When only nitrogen fixation, but not nodule initiation, is abolished in response to a local intermediate nitrate concentration, plants might be able to increase nitrogen-fixation rates more swiftly when nitrate levels drop below a certain threshold.

Consistent with our observations, it was previously reported that root hair deformation is not affected by the application of nitrate [22,53]. However, in contrast to what we found, experiments on *L. japonicus* showed that *NIN* induction is affected by the application of 10 mM NO_3_^−^ [23]. The reason for this discrepancy is unclear; a possible explanation is the timepoint at which *NIN* expression is determined. In our experiments, *NIN* expression was determined at 3 h after LCO exposure—a timepoint that precedes the initiation of the first cell divisions [16,54]. In the *L. japonicus* experiments, *NIN* expression was determined after 24 h of LCO treatment or rhizobia inoculation [23]. It is conceivable that at the 24-h timepoint cell divisions associated with nodule primordium formation have already been initiated. As *NIN* is also expressed in the early nodule primordia, it is possible that the reduction in *NIN* expression observed in *L. japonicus* in response to nitrate application results from an inhibition of nodule initiation by nitrate, rather than a direct effect of nitrate on the induction of *NIN* by LCO signalling. 

### 4.2. Nitrate Interferes with Pericycle and Cortical Cell Divisions in an Ethylene-Dependent Manner

It has been shown that induction of cell divisions associated with the formation of nodule primordia is a cytokinin–mediated process [11,12]. We show that nitrate treatment does not affect the basal cytokinin levels at any of the tested nitrate concentrations. Additionally, we show that although LCO-induced cytokinin accumulation occurs even at the highest nitrate concentration, the magnitude of this response is reduced compared to that at low nitrate levels. Although this is only statistically significant for tZ and tZR, a similar (non-significant) trend can be observed for iP and iPR; this correlates to some extent with the observed reduction in symbiotic cell divisions when nitrate concentrations are increased. However, despite a significant increase in cytokinin levels after LCO exposure at 16 mM NO_3_^−^, rhizobium-induced cell divisions were not observed at this nitrate concentration. This could indicate that (1) cell divisions leading to the initiation of nodule primordia are actively blocked, or (2) that the levels of cytokinin accumulation under these conditions do not reach the required threshold, and are therefore insufficient to initiate and sustain cell division.

Previous studies suggested a role for ethylene in the regulation of nodulation by nitrate [24,25,55]. For example, application of AVG to alfalfa roots was shown to create partial resistance to nitrate inhibition of nodulation [24]. We show that, in *M.*
*truncatula*, AVG application reduces ACC concentrations in the root susceptible zone by ~50%, which is sufficient to allow nodule formation at 16 mM NO_3_^−^. This strongly suggests that ethylene is involved in the inhibition of nodulation by nitrate. Consistently, the ethylene-insensitive *sickle* mutant forms a large number of nodules at all tested nitrate concentrations. However, a twofold reduction in nodule numbers is still observed in this mutant between 1 and 2 mM NO_3_^−^. This suggests that ethylene is not the only factor involved in the inhibition of nodulation by nitrate. Indeed, it has been shown that the autoregulation of nodulation (AON) pathway also contributes to the inhibition of nodulation by exogenous nitrate [23,56,57,58]. Genetic studies on *L. japonicus* revealed a role of HAR1 in controlling sensitivity to exogenous sources of fixed forms of nitrogen. Whereas, in wild-type plants, exogenous KNO_3_ inhibits LCO-induced expression of *NIN*, this inhibition is not observed in a *har1* knockout mutant [23,59,60]. HAR1/SUNN encodes a CLV1-type receptor kinase involved in AON [59,60]—a mechanism that restricts nodule numbers based on CLE peptides released from already existing nodules [56,57,61,62]. Mutants in this pathway are hypernodulating, and appear to be partially resistant to nitrogen inhibition of nodulation [23]. The twofold reduction in nodule numbers observed in the *sickle* mutant under elevated nitrate levels suggests that AON might work in parallel to the ethylene regulation reported here. 

## 5. Conclusions

In conclusion, our data indicate that elevated levels of nitrate interfere with the LCO-induced accumulation of cytokinins in the root susceptible zone. The inhibitory effect of nitrate can be overcome by interfering with the perception or biosynthesis of the gaseous plant hormone ethylene, suggesting a role for ethylene in the inhibition of nodulation by exogenous sources of nitrate.

## Figures and Tables

**Figure 1 genes-12-00988-f001:**
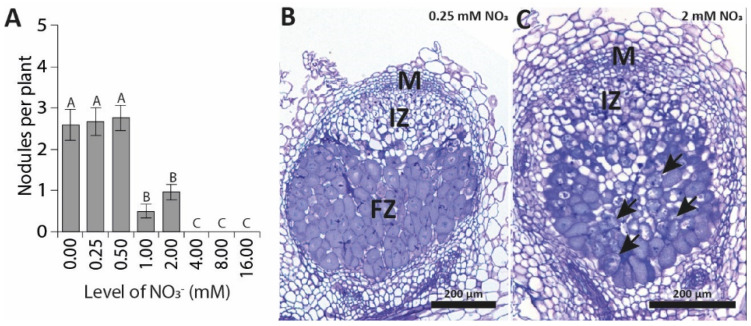
The effects of nitrate on nodulation in *M.*
*truncatula* on Fåhraeus plates: (**A**) nodule numbers per plant at 7 DPI (days post-inoculation). Representative pictures of (**B**) a nodule formed at 14 DPI on 0.25 mM NO_3_^−^, and (**C**) a nodule formed at 14 DPI on 2 mM NO_3_^−^. M: meristem; IZ: infection zone; FZ: fixation zone; arrows indicate senescing cells. Bars represent means ± SE. Bars with different letters differ significantly (*p* < 0.05).

**Figure 2 genes-12-00988-f002:**
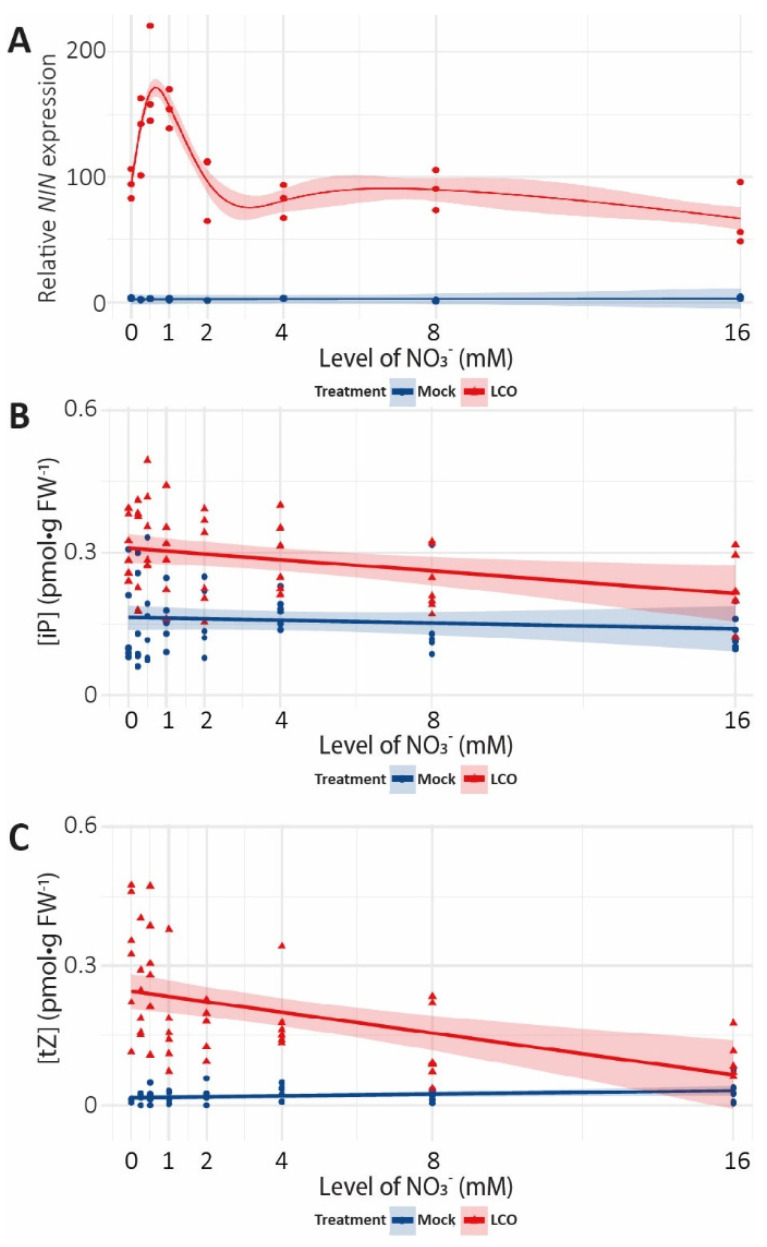
The effect of increasing levels of nitrate on LCO-induced *NIN* expression and cytokinin accumulation in the root susceptible zone of wild-type *M. truncatula* treated with mock or LCO for 3 h. (**A**) Relative *NIN* expression (*n* = 3); red (LCO) and blue (mock) dots indicate individual observations (statistics Appendix A). (**B**,**C**) Concentrations of (**B**) iP and (**C**) tZ were measured per gram of fresh weight using UPLC–MS/MS (*n* = 6); red triangles (LCO) and blue dots (mock) indicate individual observations. (**A**–**C**) Coloured line indicates average; coloured shading indicates 95% confidence interval (statistics; Appendix A).

**Figure 3 genes-12-00988-f003:**
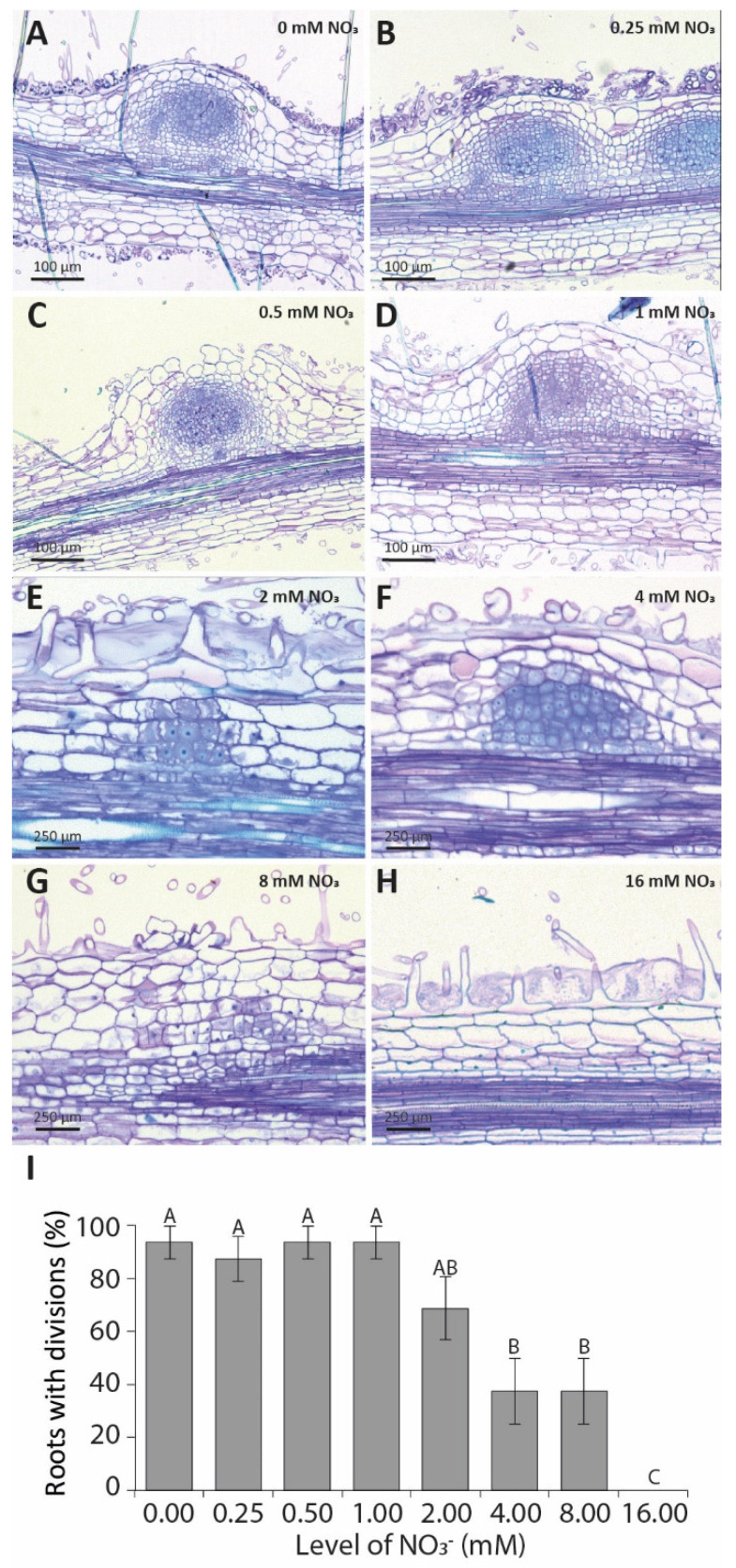
Nodulation status of the root susceptible zone after a 4-day spot inoculation with *S. meliloti*. (**A**–**H**) Morphology of nodule primordia at increasing levels of nitrate. Pictures are representative of 10 biological replicates. (**I**) The percentage of roots showing any divisions in the pericycle and/or inner cortex under increasing levels of nitrate at 4 DPI (days post-inoculation, *n* = 10). Bars represent means ± SE. Bars labelled with different letters differ significantly (*p* < 0.05).

**Figure 4 genes-12-00988-f004:**
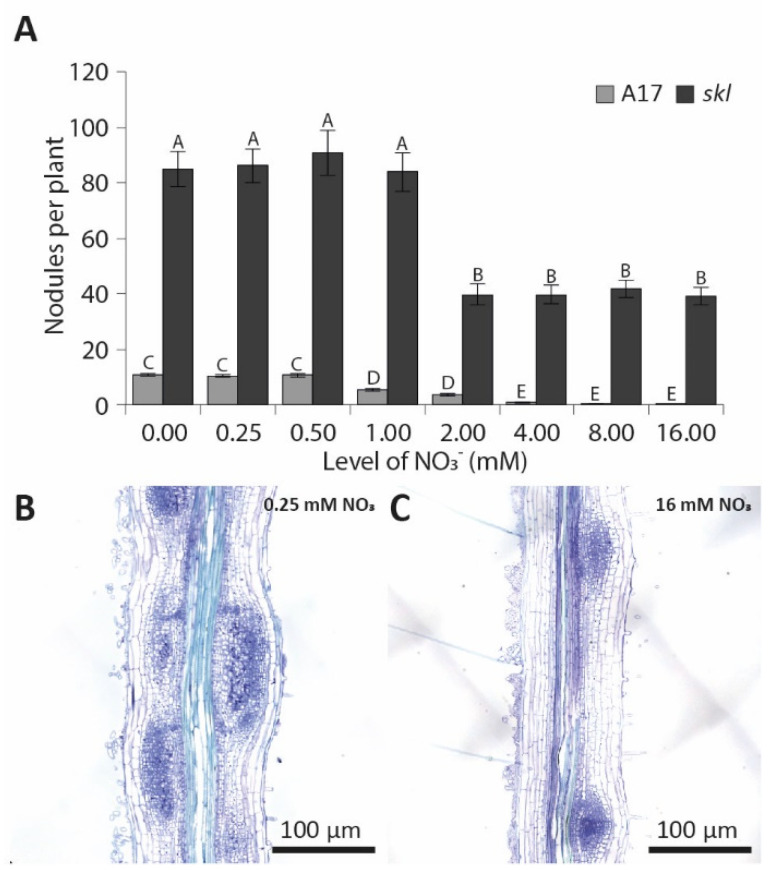
The effect of increasing levels of nitrate on nodulation in wild-type *M. truncatula* A17 and the *Mtein2*/*sickle* (*skl*) mutant. (**A**) Effect of increasing levels of nitrate on nodulation at 7 DPI (days post-inoculation, *n* = 40). Nodulation phenotype of the *Mtein2*/*sickle* mutant at (**B**) 0.25 and (**C**) 16 mM NO_3_^−^ at 4 DPI. Pictures are representative of 10 biological replicates.

**Figure 5 genes-12-00988-f005:**
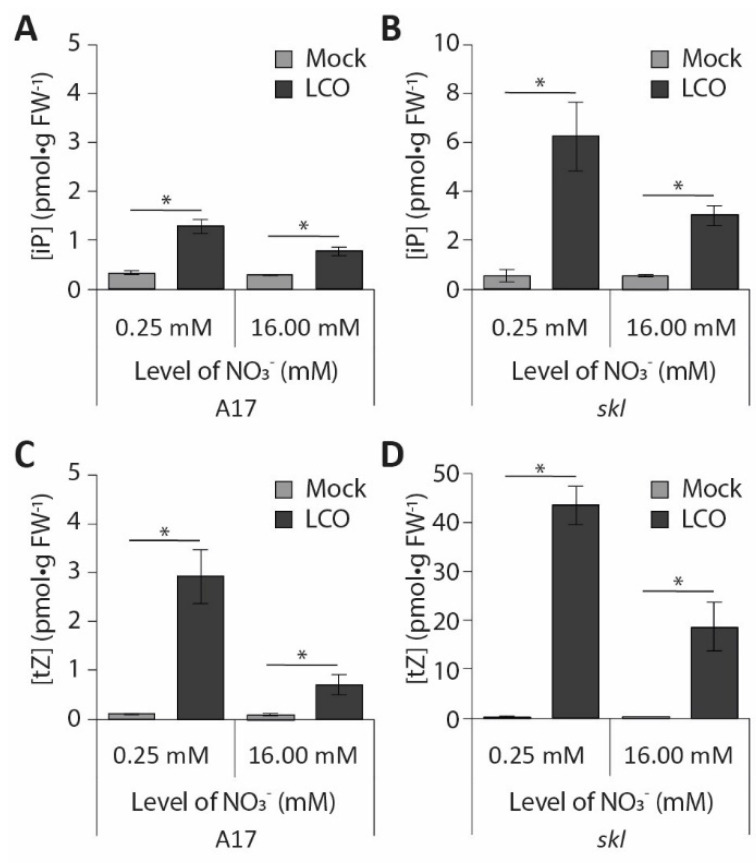
The effect of high nitrate concentration and ethylene insensitivity on LCO-induced cytokinin accumulation. Concentrations of (**A**,**B**) iP and (**C**,**D**) tZ were measured per gram of fresh weight using UPLC–MS/MS (*n* = 6) in samples taken from the root susceptible zone, and treated with mock or LCO for 3 h in (**A**,**C**) wild type or (**B**,**D**) *M. truncatula ein2*/*sickle* (*skl*). Bars represent means ± SE. Bars labelled with an asterisk (*) differ significantly (*p* < 0.05).

**Figure 6 genes-12-00988-f006:**
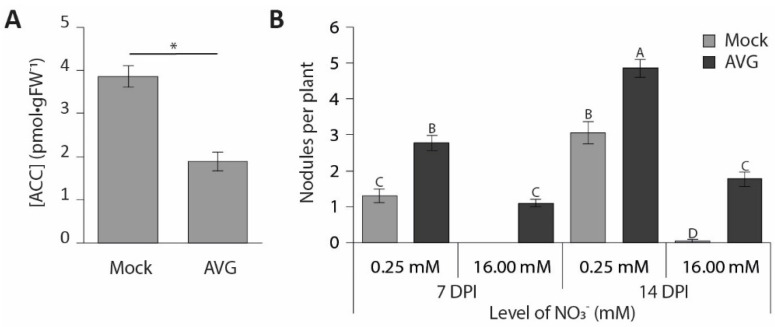
The effect of AVG application on the *M.*
*truncatula* root. (**A**) ACC measurements in the *M.*
*truncatula* root susceptible zone after 7 days of AVG application (*n* = 6). (**B**) Effect of AVG application on nodulation at 0.25 and 16 mM NO^3−^ (*n* = 40). Bars represent means ± SE. Bars labelled with different letters or an asterisk (*) differ significantly (*p* < 0.05).

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
