# Peer review of "The Effect of Exogenous Nitrate on LCO Signalling, Cytokinin Accumulation, and Nodule Initiation in Medicago truncatula"

_genes, 2021, doi:10.3390/genes12070988_

Round 1

Reviewer 1 Report

The authors have provided a concise and well-written manuscript, investigating and addressing a question in a systematic experimental manner. I have only minor comments and suggestions to improvements as listed below

Abstract

Line 19- would prefer to not use abbreviation of LCO but rather full name.

Materials and Methods

Line 107- Nodulation to square 107 (12cmx12cm) plates with different nitrate concentrations- something is missing in this sentence?

Line 115- Use full name of the organism when first used in the manuscript

Line 143- repeated words- “concentration of”

Line 145- on a shaker

Line 149- Pellets?

Line 133- Why is supplemental table S3 before S1?

Line 180-Similar to above, why is supplemental table S4 before S1 and S2 on lines 191 and line 194 respectively

Results

Line 206- Supplemental Fig S1- description is needed in the figure, Figures should be self-explanatory with all details in the legend

Line 214-215 The root length was pre-determined before the bacterial inoculation. No effect of nitrate was observed at this timepoint (five days after germination (DAG)) (Supplemental Figure S2b), indicating…………….

Line 217- S. meliloti can be used here if full name is mentioned in line 115

Line 230- well written section

Line279- Double ending dots to a sentence

Line 285-289- this is interesting

Line 306- Fig 4a?

Discussion

Line 356- any other study that has done similar work and got similar results? Can authors please give more references to this section

Line 381- references of different format

Line 433- please check the consistency in references list

Reviewer 2 Report

This study has examined the interaction of nitrate with NFs and cytokinins in order to elucidate the mechanisms underlying the inhibition of legume nodulation by fixed (chemical) forms of N. It is concluded that ethylene is involved in the inhibition of nodule initiation in Medicago by nitrate, most likely mediated by cytokinins, but also that other factors must also be involved, such as AON. The experiments have been clearly and logically performed, the data are good, and the paper is well written. It is a nice contribution to our understanding of how nitrate inhibits nodulation and I look forward to it being published.

I have some minor suggestions below:

Introduction (or Discussion). You should mention, at least in passing, that (a) nodulation by flooding-tolerant legumes like Sesbania is not inhibited by ethylene (D’Haeze et al. 2003), and (b) some dalbergioid legumes, like Arachis (groundnut/peanut) and some species in the flooding-tolerant genus Aeschynomene do not use NFs for nodulation (Giraud et al. 2007; Guha et al. 2016); what would you expect from them in terms of a nitrate response and cytokinins?

Lines 403-407. I think that you could expand slightly your discussion re. AON and N-regulation of nodulation – see recent review by Chaulagain & Frugoli (2021)

References:

Chaulagain & Frugoli (2021) Int. J. Mol. Sci. 22: 1117 https://doi.org/10.3390/ijms22031117

D’Haeze et al. (2003) PNAS 100: 11789 –11794.

Guha et al. (2016) Environmental Microbiology 18: 2575–2590.

Reviewer 3 Report

The paper requires extensive language editing especially in materials and methods section. There are several grammar mistakes, vague statements, wrong use of tenses, incorrect use/omission of articles and lack of proper punctuation. The references are inadequate and some statements made need to be backed up by citations. Follow the sections and generally stop mixing results with methods and discussion with results. I have highlighted some in the individual sections I suggest changing the sub-tittles in the results. The choice of sub-topics should not be conclusive but be more neutral rather than summarizing the results. Specific comments L92: remove preposition “for” and do not capitalize P in petri dishes L105: a comma is missing L107: the sentence is vague L111: state the use of the root samples collected L127: check the tense L130: Experiments were conducted …… to the manufacturer’s……. L139: Begin the sentence with an article L143: Remove the repeated phrase L144: Incorrect article used L191: check the chemical formula L200-212: This is the result section but it has a lot of methodology L224: instead of explaining why the white nodules were observed, just state the colour of nodules in 0-0.25 mM so that the difference just stands out. L237: I suggest a change in the sub-title to just state the kind of investigation done i.e. Effect of exogenous nitrate on root hair (development) L350: Here you should now explain the differences between pink and white nodules and probably give a citation L362-365: Is there evidence indicating that non-fixing (white) nodules regenerate the underdeveloped and dead zones as explained in lines 228-250? L281: Check the citation format L383-387: It is okay to highlight results in this section in order to discuss but do not cover the results extensively

Round 2

Reviewer 3 Report

All my comments from the first round review have been addressed by the authors.